# Influence of Citrus Scion/Rootstock Genotypes on Arbuscular Mycorrhizal Community Composition under Controlled Environment Condition

**DOI:** 10.3390/plants9070901

**Published:** 2020-07-16

**Authors:** Fang Song, Fuxi Bai, Juanjuan Wang, Liming Wu, Yingchun Jiang, Zhiyong Pan

**Affiliations:** 1Institute of Fruit and Tea, Hubei Academy of Agricultural Sciences, Wuhan 430064, China; baifuxi_kiwi@163.com (F.B.); wuliming2005@126.com (L.W.); jyc6512@aliyun.com (Y.J.); 2Institute National Agro-Technical Extension and Service Center (NATESC), Ministry of Agriculture, Beijing 100000, China; wangjuanjuan@agri.gov.cn; 3College of Horticulture and Forestry Sciences, Huazhong Agricultural University, Wuhan 430070, China; zypan@mail.hzau.edu.cn

**Keywords:** AMF community composition, citrus, scion/rootstock genotype, Illumina Miseq sequencing, PCoA.

## Abstract

Citrus is vegetatively propagated by grafting for commercial production, and most rootstock cultivars of citrus have scarce root hairs, thus heavily relying on mutualistic symbiosis with arbuscular mycorrhizal fungi (AMF) for mineral nutrient uptake. However, the AMF community composition, and its differences under different citrus scion/rootstock genotypes, were largely unknown. In this study, we investigated the citrus root-associated AMF diversity and richness, and assessed the influence of citrus scion/rootstock genotypes on the AMF community composition in a controlled condition, in order to exclude interferences from environmental factors and agricultural practices. As a result, a total of 613,408 Glomeromycota tags were detected in the citrus roots, and 46 AMF species were annotated against the MAARJAM database. Of these, 39 species belonged to *Glomus*, indicating a dominant role of the *Glomus* AMF in the symbiosis with citrus. PCoA analysis indicated that the AMF community’s composition was significantly impacted by both citrus scion and rootstock genotypes, but total samples were clustered according to rootstock genotype rather than scion genotype. In addition, AMF α diversity was significantly affected merely by rootstock genotype. Thus, rootstock genotype might exert a greater impact on the AMF community than scion genotype. Taken together, this study provides a comprehensive insight into the AMF community in juvenile citrus plants, and reveals the important effects of citrus genotype on AMF community composition.

## 1. Introduction

Arbuscular mycorrhizal symbiosis (AMS), a widespread mutualistic relationship between more than 72% of vascular plants and arbuscular mycorrhizal fungi (AMF), dates back to more than 400 million years ago [1]. All AMF belong to the phylum of Glomeromycota, which contains about 288 described species or 1700 putative species [2]. AMF can facilitate the uptake of water and mineral nutrients, such as phosphorus and nitrogen, from soil through the hypha network, and thus it can promote plant growth and development, and its tolerance to biotic/abiotic stress [3,4,5,6,7]. In exchange, the plant provides photosynthate, such as lipids, to fungi to complete the fungal life cycle [8,9,10]. Citrus roots have scarce root hairs, thus they heavily rely on AMF for absorbing mineral nutrients from soil [11]. Therefore, controlling AMF communities colonized in roots has been considered as an important measure in order to promote the sustainable development of the citrus industry by reducing the application of fertilizer [12,13].

Considering the significant roles of AMF in the ecosystem, the compositions of AMF communities have been determined through high-throughput DNA sequencing technologies in various natural and agricultural environments, including plateau, desert, forestland, grassland, shrubland and farmland [14,15,16,17,18,19,20]. According to the previous studies, AMF community composition is mainly affected by spatial scales, soil properties, climates and agricultural practices [18,20,21,22,23]. In addition to these abiotic factors, the AMF community is also impacted by host plants. Because the contributions made by different mycorrhiza to plants are varied [24,25], the host plants are able to shape the AMF communities with their functional traits [26,27], and the effects of the host plants on the AMF communities could be observed even at the genotype level [28].

Grafting is a classic technique that joins the rootstock of one plant to the scion of another, and it plays a significant role in increasing the yield, fruit quality, abiotic stress resistance and pathogen resistance of citrus plants [29,30]. Although some pioneering studies have investigated the effects of scion and rootstock genotypes on the bacterial community in grafted plants [31,32], the research on their effect on the AMF community is still limited. One of our previous studies has reported that the citrus root-associated AMF community is mainly affected by habitats, but citrus genotypes also played a minor role in shaping the AMF community [33]. However, most existing studies were conducted in field conditions, and it was difficult to exclude the huge interferences of the environment and agricultural practices.

In this study, citrus plants with different scion/rootstock genotype combinations were planted in the same field in the greenhouse, in order to exclude the interference from environment factors and all the agricultural practices (e.g., irrigation, fertilization, shoot cutting and weeding). Based on high-throughput sequencing of the 18S ribosomal RNA (rRNA) gene fragments of AMF amplified from the DNA samples prepared from citrus roots, we examined the citrus root-associated AMF community composition in a greenhouse, as well as its differences under different scion genotypes and rootstock genotypes.

## 2. Results

### 2.1. Overall Citrus AMF Taxonomic Richness

In this study, the 18S SSU rRNA (a small subunit region of ribosomal RNA) gene fragments of 18 root DNA samples, from six different scion/rootstock genotype plants grown in the same field, were investigated to reveal the effects of citrus scion/rootstock genotypes on the AMF community in a controlled condition. MISeq high-throughput sequencing indicated that a total of 1,440,324 raw reads were produced, 1,278,760 reads of which were considered as clean reads after screening. These two pair-end clean reads were then assembled into 630,728 clean tags, of which 614,590 clean tags could be assigned to the sequences deposited in the SILVA database (v108, [34]). These clean tags could be further divided into six distinct taxonomic groups based on the amplified SSU rRNA gene tags (Table 1). As expected, most tags amplified with the primer pair AMV4.5NF/AMDGR (613,408 tags, accounting for 99.81% of the total tags) belonged to Glomeramycota, which were also considered to be arbuscular mycorrhizal fungi (AMF). Some non-AMF sequences were also detected in this study, including Basidiomycota (794 tags, accounting for 0.13% of the total tags), Ascomycota (41, 0.01%) and Chytridiomycota (13, 0.00%). A small proportion of the tags (334, 0.05%) could not be unambiguously assigned to any of the above phyla (Table 1). To better clarify the genetic diversity of the AMF community associated with the citrus root, 614,590 clean tags were further clustered into 132 operational taxonomic units (OTUs), based on 97% sequence similarity. The 87 OTUs (accounting for 65.91% of the total OTUs) belonged to Glomeramycota, followed by Basidiomycota (13, 9.85%), Ascomycota (8, 6.06%), Chytridiomycota (1, 1.76%), and the remaining 23 OTUs (17.42%) belonged to the unclassified group (Table 1).

To clearly define the AMF community compositions associated with citrus roots at the species level, the clean tags were aligned to the sequences deposited in the MAARJAM database [35], which contains all the published Glomeromycota SSU rRNA gene sequences. A total of 475,917 tags (75.46% of the total clean tags) from 18 citrus root samples were annotated to 46 virtual taxa (VT, an AMF molecular species) according to the MAARJAM database (Appendix A). These 46 VT represented a high level of AMF diversity, associated with the citrus roots in a single field in a greenhouse. These 46 VT could be further divided into four families (Table 2), including Glomeraceae (39 VT; according to 474,991 clean tags, 99.81% of the 475,917 clean tags assigned to MAARJAM database), Paraglomeraceae (3VT, 832 tags), Claroideoglomeraceae (3 VT, 88 tags) and Gigasporaceae (1VT, 6 tags). Obviously, *Glomus* was the dominant AMF family associated with citrus roots, which could be further divided into three subgroups, including *Glomus* group A, *Glomus* group B and *Glomus* group C (Figure 1). Notably, the three most abundant AMF species, namely, Glomus.Yamato08.A1_VTX00100 (134,970 tags, according to 28.36% of the 475,917 clean tags assigned to the MAARJAM database), Glomus.acnaGlo2_VTX00155 (71,418, 15.01%) and Glomus.Yamato09.A2_VTX00248 (66,821, 14.04%), were the dominant AMF species associated with citrus roots.

### 2.2. Difference in α Diversity of AMF Community among Different Citrus Genotypes

To explore the effects of citrus genotypes on the compositions of the AMF communities associated with citrus roots, the α diversity (including species richness and genetic diversity) of the AMF communities from citrus roots under six different combinations of citrus scion/rootstock genotypes was assessed according to four different indices, namely, the Sobs index and Ace index (for the estimation of species richness), and the Simpson index and Shannon Index (for the estimation of genetic diversity). As shown in Table 3, the AMF richness of *Orange/Poncirus* was the highest, with the maximal Sobs and Ace values, and those of *Orange/CitrangeCitrange* were the lowest. Otherwise, the AMF diversity of different citrus genotypes was nearly in line with the AMF richness. It should be noted that a low value for the Simpson index meant a high diversity of a given AMF community, and that the Simpson value was the opposite of the Shannon value. *Orange/Poncirus* exhibited the highest AMF diversity, followed by *Poncirus*/*Poncirus*, *Mandarin*/*Poncirus* and *Pummelo*/*Poncirus*, whereas *Orange/Citrange* and *Citrange/Citrange* displayed the lowest AMF diversity. Interestingly, significant differences in both AMF richness and AMF diversity were observed between *Orange/Citrange* and *Orange/Poncirus.* These results indicated the potential role of rootstock genotypes in regulating the citrus root-associated AMF community richness and diversity. Additionally, the rarefaction curves were developed according to Sobs Index, in order to assess the sequencing depth of all 18 samples. As shown in Appendix A, most of the rarefaction curves reached a plateau, indicating that the sequencing depth was sufficient for AMF richness and diversity assessment.

### 2.3. Citrus Genotypes Shape the AMF Community Composition

Principal coordinates analysis (PCoA) was performed based on the OTU abundance of the entire set of citrus root samples, in order to further explore the differences in AMF community composition among the different citrus genotypes. The PCoA analysis identified two components accounting for 62.14% of the total variance, which was explained by axis 1 (47.06%) and by axis 2 (15.08%), respectively (Figure 2). The PCoA analysis results revealed that the differences in AMF community composition among all the citrus root samples could be attributed to the genotype of the citrus rootstock. To be more specific, the AMF communities of *Citrange/Citrange* and *Orange/Citrange* obviously clustered to the right, with the same rootstock genotype, and the AMF communities of *Mandarin/Poncirus*, *Orange/Poncirus* and *Pummelo/Poncirus* showed clear clustering to the left with the same rootstock genotype. This result revealed that rootstock genotype played a major role in shaping the AMF community composition. To further classify the effects of the citrus scion/rootstock genotype on the AMF community’s composition, we focused on two particular scion/rootstock genotype combinations: (1) *Orange/Poncirus* and *Orange/Citrange*, and (2) *Orange/Citrange* and *Citrange/Citrange*. As shown in Figure 2, Orange/Poncirus (Yellow) and *Orange/Citrange* (Blue), with the same scion of Newhall sweet orange grafted onto different rootstocks of *Poncirus* and *Citrange*, were clearly separated into two groups in both PC1 and PC2 axes. However, *Orange/Citrange* (Blue) and *Citrange/Citrange* (Red), with different scions of Newhall sweet orange and *Citrange* grafted onto the same rootstock of *Citrange*, were clustered together. Thus, rootstock genotype might exert a greater effect on AMF community composition than the scion genotype.

To further evaluate the effect of scion genotype on the AMF community, we performed PCoA analyses of the AMF community from *Mandarin/Poncirus*, *Orange/Poncirus*, *Pummelo/Poncirus* and *Poncirus/Poncirus*, with four different scions of *Mandarin*, *Newhall sweet orange*, *Pummelo* and *Poncirus* grafted onto the same rootstock of *Poncirus*. As shown in Figure 3, the examined samples with four different scions grafted onto the same rootstock of *Poncirus* were clustered into different groups, which accounted for 53.63% of the total variance in axis 1 (44.36%) and axis 2 (9.27%) (Figure 3). To be more specific, samples of *Poncirus/Poncirus* (Red) were clustered in the top left, those of *Mandarin/Poncirus* (Orange) were clustered in the top right, while those of *Orange/Poncirus* (Green) and *Pummelo/Poncirus* (Blue) were clustered together in the bottom. These results suggested that both the scion and rootstock genotypes were able to shape the AMF community composition of the citrus roots, and that the rootstock might exert a greater impact on AMF diversity than the scion. 

### 2.4. Relative Abundance of AMF Species under Different Citrus Genotypes

The difference in AMF community composition between six citrus scion/rootstock genotypes could be attributed to the diversity and richness of the AMF species. We performed a network analysis of the AMF species under different citrus genotypes. The results indicated that of all the 46 AMF species, 21 (45.65%) AMF species were found to be present in all the six samples of different citrus genotypes (Figure 4). In addition, six AMF species (13.04%) were present in only one sample. We also found that Glomus.MO-G16_VTX00072 was only detected in the roots of *Citrange/Citrange*, that Glomus.Wirsel.OTU14_VTX00137 and Paraglomus.Alguacil12b.ACA1_VTX00352 were only detected in the roots of *Pummelo*/*Poncirus*, and that Glomus.MO-G22_VTX00125, Glomus.sp._VTX00420, Glomus.MO-G23_VTX00222 and Glomus.Wirsel.OTU6_VTX00202 were only detected in the roots of *Orange*/*Poncirus*.

Statistical analysis indicated significant differences in the relative abundance of 15 non-specific AMF species under six different scion/rootstock genotypes (Table 4). The relative abundance of Glomus.Yamato08.A1_VTX00100 was found to be significantly higher in the samples with the same rootstock of *Citrange* (i.e., *Citrange/Citrange* and *Orange/Citrange*) than in the samples with rootstock of *Poncirus* (*Poncirus*/*Poncirus*, *Pummelo/Poncirus*, *Mandarin/Poncirus* and *Orange/Poncirus*). In contrast, the relative abundance of Glomus.Yamato09.A2_VTX00248 was significantly higher in the samples with the rootstock of *Poncirus* than in the samples with the rootstock of *Citrange*. Additionally, Glomus.MO-G44_VTX00410 and Glomus.NF05_VTX00322 showed a significant enrichment in *Orange/Poncirus* compared to the other three samples with the same rootstock but different scions.

## 3. Discussion

Despite the important role of arbuscular mycorrhizal fungi (AMF) in mineral nutrient uptake for citrus plants, our understanding of the relationship between AMF and citrus remains relatively limited. In this study, we investigated the AMF community colonized in the citrus roots of six different genotypes in the greenhouse, including *Poncirus*/*Poncirus*, *Citrange/Citrange*, *Mandarin*/*Poncirus*, *Pummelo*/*Poncirus*, *Orange/Poncirus* and *Orange/Citrange*. We found that the genus *Glomus* dominated all the six samples, indicating the absolute dominance of *Glomus* in symbiosis with the citrus root across different citrus genotypes. Furthermore, we also revealed that the AMF community composition was affected by the citrus scion/rootstock genotype, and that the rootstock genotype of citrus might have a greater effect than the scion genotype. 

To explore the effects of citrus genotypes on AMF community composition, a total of 18 root DNA samples from citrus plants grown under the same controlled condition, under six different scion/rootstock genotypes, were prepared and used for PCR amplification of the 18S small subunit (SSU) rRNA gene fragment, using the highly AMF-specific primer pair AMV4.5NF/AMDGR. In this study, a total of 1,440,324 raw reads were yielded from 18 root DNA samples, of which 1,278,760 (88.78%) were clear reads, and 46 VT (AMF species) were annotated. In the past few decades, the identification of citrus AMF species has been performed based on spore morphology [36,37]. The morphological analysis indicated that 18 AMF species belonging to five families, including Glomeraceae (9 species accounting for 50%), Acaulosporaceae (4 species), Claroideoglomeraceae (2 species), Gigasporaceae (1 species) and Pacisporaceae (2 species), have been detected from the rhizosphere soil of citrus [38]. However, our study results indicated that almost 99.81% of the tags and 84.78% of the VT belonged to *Glomus* (the only genus of Glomeraceae), indicating an absolute dominance of the genus *Glomus*. This phenomenon also revealed that the AMF species of spores in the rhizosphere soil were largely different from those colonizing in the citrus roots [39,40], which further implied that the selectivity of the plant species played an important role in the colonizing AMF community [41]. As described in previous reports, host plant species has been identified as an important factor in shaping the root-colonizing AMF community’s composition [42,43]. Thus, *Glomus* AMF species might be recruited by citrus plants to colonize in the root cortical cells. In addition, about half of the AMF species (21 VT) were present in all the six samples of different citrus genotypes, while only six AMF species existed in a single sample, which might be attributed to the fact that all the six samples were planted in the same field in the greenhouse. Habitats were reported to have a major effect on the AMF community [44], thus the native AMF species of the *Glomus* genus were rapidly colonized in all the citrus seedlings.

Citrus roots have scarce root hairs, and thus they heavily rely on AMF for mineral nutrient uptake. Considering the significant role of AMF in citrus production, many studies focused on the physiological interaction and molecular signaling dialogue between root-colonizing AMF and citrus plants [45,46,47]. However, whether and to what degree the citrus genotype could affect the AMF community remains unclear. Our previous study indicated that the AMF community is dominantly impacted by habitat, and that citrus genotype might have a slight, but significant, effect on the AMF community composition [33. However, all the samples in the previous studies were collected from different citrus production areas in south China, and the AMF community was influenced by many field environment factors, such as soil properties, illumination, humidity, diseases and agricultural practices. Thus, in this study, we investigated the impact of the citrus scion/rootstock genotype on AMF community in a particular field in a greenhouse, with the same agricultural management, in order to exclude the interference of environment factors and agricultural practices. As was expected, the majority of samples were clustered based on different rootstock genotypes (accounting for 62.14% of the total variance, Figure 2), indicating the dominant role of citrus rootstock genotype in regulating AMF community composition. Interestingly, the further PCoA results indicated that both the scion and rootstock genotypes had a significant impact on the AMF community composition, and that the rootstock genotype might exert a greater impact than the scion genotype (Figure 2 and Figure 3). In addition, out of six different scion/rootstock combinations, only *Orange/Citrange* and *Orange/Poncirus*, which were obtained by grafting scion *Newhall sweet orange* onto rootstocks *Citrange* and *Poncirus*, respectively, exhibited a significant difference in the α diversity (Table 3). This result indicates the potential role of rootstock genotype in regulating the community richness and diversity of AMF associated with citrus roots.

Citrus is a woody perennial plant, and it is widely planted all over the world in a form of graft. For instance, the commercial citrus cultivars *Newhall sweet orange* and *Mandarin* as scion are grafted onto the rootstocks *Poncirus* and *Citrange*, which is of significant importance for increasing the fruit’s quality and its resistance to multiple biotic and abiotic stresses [48,49,50,51,52]. However, how scion and rootstock combinations affect the AMF community in citrus roots is still poorly understood. The previous study reported that the scion could influence the microbe’s symbiosis with the rootstock through the primary and secondary metabolites [8,9,53,54], phytohormones [55], or some regulatory factors [56,57]. This study also revealed the significant impact of citrus scion genotype on the root-associated AMF community. It was reported that AMF inhabited the roots, and formed a mutualistic symbiont in the cortical cells of the citrus roots [5,10]. Thus, it could be speculated that rootstock genotype might exert a greater impact on the AMF community than scion genotype. This result was in line with those of similar studies on the root-associated microbiome [31,58,59], but it was in opposition to those of our previous study of field condition. One possible explanation is the age of the citrus plants. In the previous study, the grafted citrus plants were more than 20 years old, and they might have modified the surrounding environment through root exudates [60,61]. However, 1-year-old juvenile citrus plants were grafted and transplanted into the greenhouse for only 1-year growth. The photosynthates synthesized from the scion (above ground part) of juvenile citrus plants were very limited [62], and less photosynthates will be translocated to the roots, since the root functions were not well recovered after transplantation into the new environment [63]. Another possible explanation lies in the different environmental conditions in different studies. As was reported, environmental factors were the dominant regulator of the AMF community [64,65,66]. All our samples used for assessing the impact of scion/rootstock genotype on AMF community were obtained from one small field in the greenhouse, but the scales of the citrus orchards in the previous studies were much larger than the greenhouse. Thus, the environmental conditions, especially the soil properties, might interfere with the effects of citrus scion/rootstock genotype on AMF community.

## 4. Conclusions

In summary, this study comprehensively analyzed AMF community composition as associated with citrus roots in a greenhouse. The AMF genus *Glomus* was found to be dominant in all the six samples, but different scion/rootstock genotypes specifically recruited certain AMF species from the surrounding soils. Our results indicated that AMF community composition was affected by citrus scion/rootstock genotype, and that the rootstock genotype of citrus might play a greater role than scion genotype. Further studies are needed in order to better understand the mechanisms by which grafted citrus plants shape the AMF community, and an investigation into the role of their metabolites or regulatory factors in shaping AMF communities is of great importance. 

## 5. Materials and Methods 

### 5.1. Plant Materials and Sample Preparation

*Poncirus* and *Citrange* seeds were generated in a sterile condition, and then transplanted into the sterile sand for one-year growth. In September of the next year, different scions were grafted onto the rootstocks. The grafted citrus plants were used for the subsequent experiment after the scions were more than 30 cm in length. Citrus plants with different genotypes were planted in the same field in the greenhouse to exclude the interference of environmental factors. A total of six scion/rootstock combinations of citrus cultivars were utilized to explore the effects of citrus genotype on AMF community composition, including (i) *Poncirus* (*Poncirus trifoliata*), grafted onto *Poncirus* and labeled *Poncirus*/*Poncirus*, (ii) *Citrange* (*Citrus sinensis*×*Poncirus trifoliata*), grafted onto *Citrange* and labeled *Citrange/Citrange*, (iii) *Mandarin* (*Citrus reticulata*), grafted onto *Poncirus* and labeled *Mandarin*/*Poncirus*, (iv) *HB pummelo* (*Citrus grandis*), grafted onto *Poncirus* and labeled *Pummelo*/*Poncirus*, (v) *Newhall sweet orange* (*Citrus sinensis*), grafted onto *Poncirus* and labeled *Orange/Poncirus*, and (vi) *Newhall sweet orange*, grafted onto *Citrange* and labeled *Orange/Citrange* (Table 5, Appendix A). After one year, a total of 18 citrus root samples were harvested for further experiments with three biological replicates for each citrus genotype. 

The root samples were processed according to a previous study with minor modifications [67]. Lateral roots were collected from the citrus samples and loose soil was removed with sterile gloves (sprayed with 70% Ethanol). To remove the tightly attached soil, lateral roots were transferred into sterile 50-mL tubes containing 25 mL phosphate buffer (per litre: 6.33 g of NaH_2_PO_4_∙H_2_O, 16.5 g of Na_2_HPO_4_∙7H_2_O, 200 µL Silwet L-77) and vortexed at a maximal speed. Buffers were refreshed until the lateral roots were totally cleaned. Subsequently, lateral roots were subjected to an ultrasonic cleaning for 10 min to remove the tiny soils and loose microbes on the root surface. After processing, lateral roots were frozen with liquid nitrogen and stored at −80 °C. 

### 5.2. DNA Extraction, Library Construction and Sequencing

Total DNA of the samples was extracted with a CTAB method reported previously, unique for citrus plants [68]. To construct the sequencing library, the 18S small subunit regions of riboso mal RNA gene (SSU rRNA) fragments were amplified with a widely used primer pair AMV4.5NF/AMDGR [69,70] fused with two sequencing adaptors and dual index sequences. PCR products were purified and measured with LabChip GX (Caliper, USA). Subsequently, the qualified libraries were uploaded to the Illumina platform (MISeq) for sequencing by Beijing Genomics Institute (BGI, Shenzhen, China). 

### 5.3. Data Analysis

After MISeq sequencing, the raw reads were screened to remove the low-quality reads, including the following: (i) the truncated sequence reads without an average quality of 20 over a 25-bp sliding window, based on the phred algorithm, and trimmed reads with less than 75% of the original length; (ii) the reads contaminated by adaptors; (iii) the reads containing an ambiguous base; and (iv) the reads with low complexity. Then the two pair-ended clean reads were assembled into a single tag with Fast Length Adjustment of Short reads (FLASH, v1.2.11) if they were overlapped [71], and the clean reads without overlaps were removed. Subsequently, tags were clustered into OTUs based on 97% sequence similarity, using USEARCH software [72]. The most abundant tag from each OTU was selected as the representative sequence, and the potential chimeras were identified and removed from the representative sequence with UCHIME [73]. 

The representative sequences were annotated to fungal OTUs against the SILVA database (v108, [34) ], and those identified as Glomeromycota sequences were further assigned to AMF species (VT, virtual taxa) against the MAARJAM database [35]. We performed a BLAST search to assign the representative sequences to the SILVA database and MAARJAM database by the following criteria: the sequence similarity was ≥97%, and the BLAST e-value was <1 × 10^−10^.

### 5.4. Phylogenetic Analysis

For phylogenetic analysis, the representative sequences of Glomeromycota VT (the most abundant sequences of each VT) detected in all the samples were aligned using the software of the multiple sequence alignment program (MAFFT, version 7) [74]. Then, the alignment results were subjected to a neighbor-joining analysis by TOPALi (v2.5), and the phylogenetic tree was developed using the F84 model with gamma substitution rates and bootstrapping over 100 runs [75].

### 5.5. Statistical Analysis 

In order to compare the diversity and richness of the AMF community under different citrus genotypes, α diversity (within-sample diversity or estimate of species richness) of different samples was measured. The AMF diversity indices (Shannon Index and Simpson Index) and richness indices (sbserved species and Ace Index) were measured with QIIME, as described in a previous study [70]. The rarefaction curve of all the detected samples was plotted based on the indices of Sobs using R software (version 2.15.3). To clearly illustrate the effect of citrus scion/rootstock genotypes on the AMF community, the similarities between the AMF communities were calculated using a Bray–Curtis method based on the OTU abundance. A matrix of Bray–Curtis distances between the entire 18 citrus root samples was created with QIIME, and a principal coordinate analysis (PCoA) of all relevant samples, and a subgroup of such samples, was performed based on the Bray–Curtis distances. The corresponding results were plotted with R software (version 2.15.3). Different dimensions of PC1, PC2 or PC3 were shown in the PCoA map according to the clustering of samples. The Fisher least significant difference (LSD) test of Statistical Product and Service Solutions (SPSS, [76]) were used for detecting the differentially abundant AMF species.

## Figures and Tables

**Figure 1 plants-09-00901-f001:**
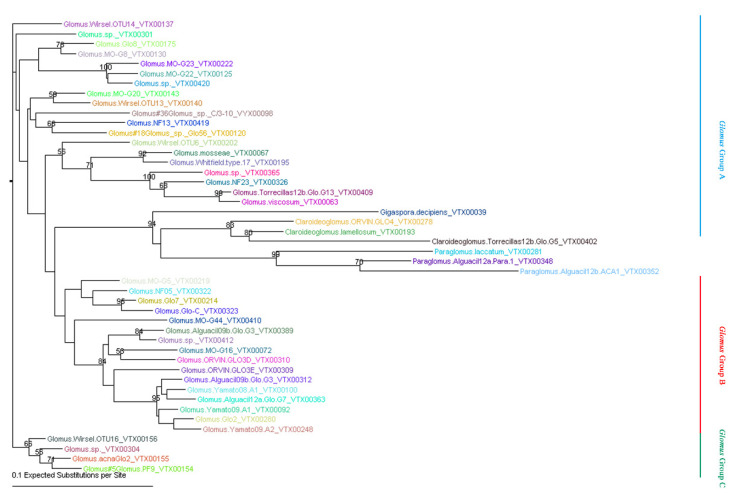
Neighbor-joining phylogenetic tree of AMF species detected by blasting against MAARJAM database. The F84+Gamma nucleotide substitution model was used and bootstrap values of > 50 are shown.

**Figure 2 plants-09-00901-f002:**
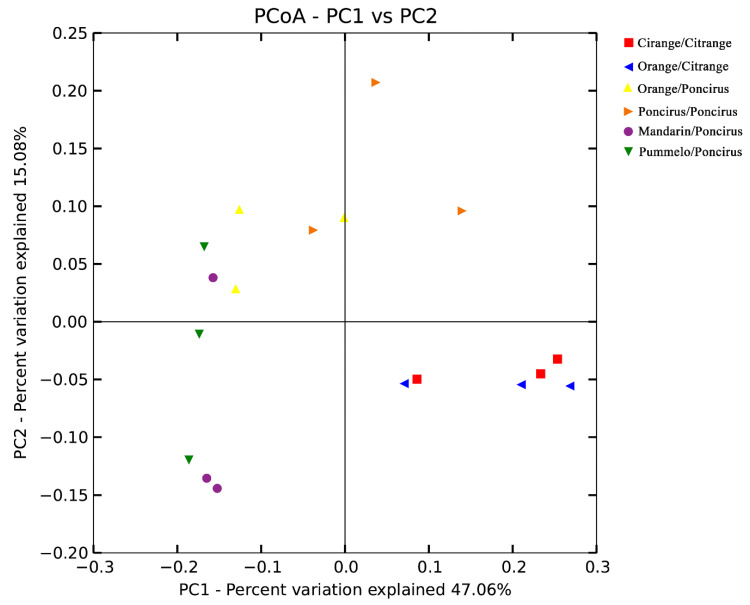
Principal coordinate analysis (PCoA) of variance in the citrus root-associated AMF community from six different scion/rootstock genotype combinations. *Poncirus*/*Poncirus* (*Poncirus* (*Poncirus trifoliata*) grafted onto *Poncirus*), *Citrange*/*Citrange* (*Citrange* (*Citrus sinensis*×*Poncirus trifoliata*) grafted onto *Citrange*), *Mandarin*/*Poncirus* (*Mandarin* (*Citrus reticulata*) grafted onto *Poncirus*), *Pummelo*/*Poncirus* (*HB pummelo* (*Citrus grandis*) grafted onto *Poncirus*), *Orange/Poncirus* (*Newhall sweet orange* (*Citrus sinensis*) grafted onto *Poncirus*) and *Orange/Citrange* (*Newhall sweet orange* grafted onto *Citrange*).

**Figure 3 plants-09-00901-f003:**
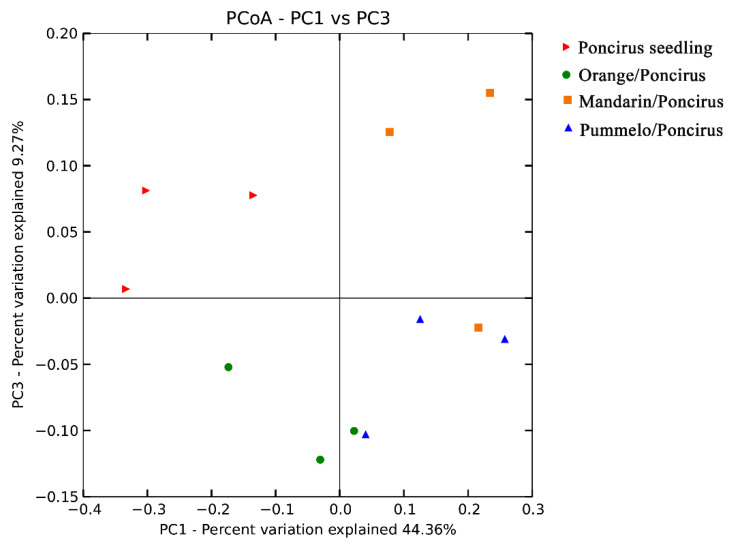
Principal coordinate analysis (PCoA) of variance in the citrus root-associated AMF communities from four grafted citrus plants with different scions grafted onto the same rootstock (Poncirus/Poncirus, Mandarin/Poncirus, Pummelo/Poncirus and Orange/Poncirus).

**Figure 4 plants-09-00901-f004:**
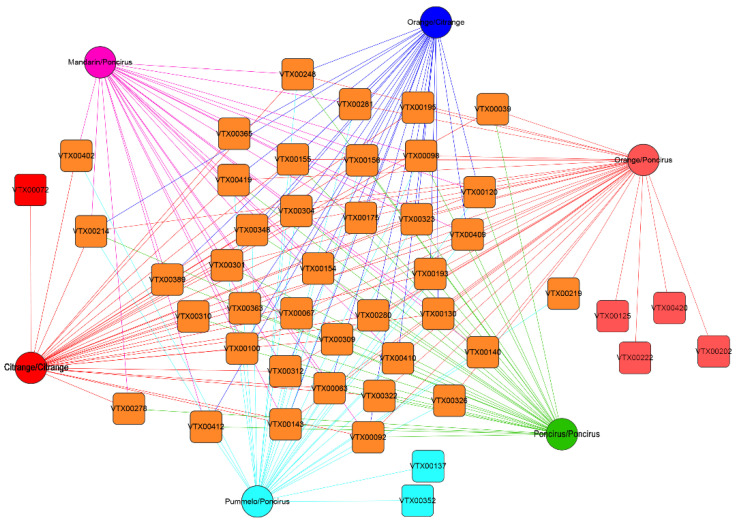
Networks analyses of all the root-associated AMF species from six different citrus scion/rootstock genotype combinations. AMF species which were specifically detected in one of the citrus scion/rootstock genotype combinations are highlighted in different colors, including *Poncirus*/*Poncirus* (Green), *Citrange*/*Citrange* (Red), *Mandarin*/*Poncirus* (Rose), *Pummelo*/*Poncirus* (Light Blue), *Orange/Poncirus* (Pink) and *Orange/Citrange* (Dark Blue).

**Table 1 plants-09-00901-t001:** Proportional distribution of total tags and generated OTUs (Operational Taxonomic Units) grouped by phyla of fungi from all citrus root samples, through blasting against the the SILVA database.

Phylum	OTUs	Tags
Glomeromycota	87 (65.91%)	613,408 (99.81%)
Basidiomycota	13 (9.85%)	794 (0.13%)
Ascomycota	8 (6.06%)	41 (0.01%)
Chytridiomycota	1 (0.76%)	13 (0.00%)
unclassified	23 (17.42%)	334 (0.05%)

**Table 2 plants-09-00901-t002:** Identification and classification of arbuscular mycorrhizal fungi (AMF) species from citrus roots in the greenhouse. AMF sequences were divided into virtual taxa through blasting against the MAARJAM database.

AMF Family	AMF Species	OTU	Tags
Glomeraceae (39 VT, corresponding to 474,991 clean reads, accounting for 99.81% of the total (475,917) clean reads against the MAARJAM database)	Glomus.Yamato08.A1_VTX00100	2	134,970
	Glomus.acnaGlo2_VTX00155	3	71,418
	Glomus.Yamato09.A2_VTX00248	1	66,821
	Glomus.viscosum_VTX00063	8	37,975
	Glomus.Glo7_VTX00214	3	30,951
	Glomus.NF13_VTX00419	2	21,670
	Glomus.Torrecillas12b.Glo.G13_VTX00409	3	19,765
	Glomus.sp._VTX00304	1	16,515
	Glomus.Wirsel.OTU16_VTX00156	1	16,510
	Glomus.Glo-C_VTX00323	2	10,763
	Glomus.sp._VTX00412	1	8217
	Glomus.Yamato09.A1_VTX00092	2	6499
	Glomus.Glo8_VTX00175	1	3781
	Glomus.MO-G5_VTX00219	1	3736
	Glomus.MO-G8_VTX00130	1	3311
	Glomus_sp._Glo56	1	3092
	Glomus.MO-G44_VTX00410	2	3038
	Glomus.Alguacil09b.Glo.G3_VTX00389	1	2478
	Glomus.MO-G23_VTX00222	1	2337
	Glomus.sp._VTX00301	2	2292
	Glomus.PF9_VTX00154	1	1567
	Glomus_sp._C/3-10	2	1305
	Glomus.ORVIN.GLO3E_VTX00309	1	1290
	Glomus.MO-G20_VTX00143	1	1279
	Glomus.Alguacil09b.Glo.G3_VTX00312	1	802
	Glomus.NF05_VTX00322	1	657
	Glomus.sp._VTX00420	1	548
	Glomus.MO-G22_VTX00125	1	404
	Glomus.Alguacil12a.Glo.G7_VTX00363	1	300
	Glomus.NF23_VTX00326	2	260
	Glomus.Wirsel.OTU13_VTX00140	1	159
	Glomus.Glo2_VTX00280	1	96
	Glomus.Wirsel.OTU6_VTX00202	1	75
	Glomus.ORVIN.GLO3D_VTX00310	1	68
	Glomus.Wirsel.OTU14_VTX00137	2	17
	Glomus.mosseae_VTX00067	1	9
	Glomus.MO-G16_VTX00072	1	6
	Glomus.sp._VTX00365	1	5
	Glomus.Whitfield.type.17_VTX00195	1	5
Claroideoglomeraceae (3 VT, 88 reads, 0.02%)	Claroideoglomus.Torrecillas12b.Glo.G5_VTX00402	2	37
	Claroideoglomus.lamellosum_VTX00193	1	37
	Claroideoglomus.ORVIN.GLO4_VTX00278	2	14
Paraglomeraceae (3 VT, 832 reads, 0.17%)	Paraglomus.Alguacil12a.Para.1_VTX00348	12	787
	Paraglomus.laccatum_VTX00281	1	37
	Paraglomus.Alguacil12b.ACA1_VTX00352	1	8
Gigasporaceae (1 VT, 6 reads, 0.00%)	Gigaspora.decipiens_VTX00039	1	6

**Table 3 plants-09-00901-t003:** α diversity of AMF identified in citrus root samples from six scion/rootstock genotypes. AMF richness is reflected by Sobs and Ace Index, and AMF diversity is reflected by Shannon Index and Simpson Index. Data are means ± SE. Different lowercase letters (i.e., a and b) indicate significant differences of *p*< 0.05.

Sample	Sobs	Ace	Shannon	Simpson
*Citrange/Citrange*	55.00 ± 2.65 ab	60.74 ± 4.56 ab	1.79 ± 0.2 b	0.34 ± 0.07 a
*Orange/* *Citrange*	48.33 ± 4.26 b	54.83 ± 3.46 b	1.80 ± 0.52 b	0.35 ± 0.16 a
*Poncirus*/*Poncirus*	55.33 ± 2.91 ab	59.37 ± 3.93 ab	2.43 ± 0.18 ab	0.15 ± 0.04 ab
*Pummelo*/*Poncirus*	56.33 ± 6.36 ab	58.56 ± 5.99 ab	2.27 ± 0.17 ab	0.16 ± 0.02 ab
*Mandarin*/*Poncirus*	58.00 ± 3.79 ab	64.47 ± 5.04 ab	2.28 ± 0.14 ab	0.15 ± 0.03 ab
*Orange/Poncirus*	63.00 ± 4.04 a	77.40 ± 12.21 a	2.64 ± 0.14 a	0.11 ± 0.02 b

**Table 4 plants-09-00901-t004:** The relative abundance of AMF species with significant differences among six citrus scion/rootstock genotypes. Data are presented as means ± SE. Different lowercase letters (i.e., a and b) indicate significant differences within the same row (*p* < 0.05).

AMF Species	*Citrange*/ *Citrange*	*Orange*/ *Citrange*	*Poncirus*/ *Poncirus*	*Pummelo*/ *Poncirus*	*Mandarin*/ *Poncirus*	*Orange*/ *Poncirus*
Glomus.MO-G44_VTX00410	0 ± 0 b	0 ± 0 b	0.05 ± 0.05 b	0.38 ± 0.38 b	0.26 ± 0.18 b	2.27 ± 1.29 a
Glomus.MO-G8_VTX00130	0.01 ± 0 b	0 ± 0 b	0.07 ± 0.05 b	0.38 ± 0.2 ab	0.47 ± 0.27 ab	2.29 ± 1.65 a
Glomus.sp._VTX00301	0.03 ± 0.01 b	0.01 ± 0.01 b	0.1 ± 0.02 b	0.74 ± 0.26 ab	1.33 ± 0.61 a	0.04 ± 0.02 b
Glomus.NF05_VTX00322	0.03 ± 0.03 b	0 ± 0 b	0.04 ± 0.03 b	0.08 ± 0.08 b	0 ± 0 b	0.49 ± 0.25 a
Paraglomus.Alguacil12a.Para.1_ VTX00348	0.04 ± 0.01 b	0.06 ± 0.01 b	0.05 ± 0.02 b	0.1 ± 0.07 b	0.4 ± 0.23 a	0.13 ± 0.04 ab
Glomus.sp._VTX00304	0.18 ± 0.06 b	0.08 ± 0.08 b	0.63 ± 0.13 ab	6.71 ± 0.48 a	8.05 ± 4.37 a	0.51 ± 0.29 ab
Glomus.Wirsel.OTU16_VTX00156	0.21 ± 0.09 b	0.08 ± 0.08 b	0.65 ± 0.16 b	5.4 ± 1.62 ab	9.29 ± 4.52 a	0.5 ± 0.26 b
Glomus.ORVIN.GLO3E_VTX00309	0.86 ± 0.33 a	0 ± 0 b	0 ± 0 b	0 ± 0 b	0 ± 0 b	0.37 ± 0.37 ab
Glomus.Torrecillas12b.Glo.G13_ VTX00409	0.91 ± 0.33 b	0.7 ± 0.26 b	0.64 ± 0.35 b	8.15 ± 5.2 a	5.83 ± 0.66 ab	3.08 ± 0.14 ab
Glomus.NF13_VTX00419	1.55 ± 1.29 ab	6.74 ± 3.42 ab	2.47 ± 0.3 ab	2.19 ± 0.79 ab	0.83 ± 0.2 b	7.79 ± 4 a
Glomus.Glo-C_VTX00323	1.59 ± 0.59 b	0.74 ± 0.40 b	8.03 ± 3.78 a	0.19 ± 0.18 b	0.02 ± 0.01 b	0.22 ± 0.19 b
Glomus.Glo7_VTX00214	3.04 ± 1.14 b	3.65 ± 2.81 b	15.18 ± 7.15 a	2.68 ± 2.23 b	0.73 ± 0.59 b	5.34 ± 3.88 ab
Glomus.Yamato09.A2_VTX00248	3.52 ± 2.41 b	2.75 ± 1.38 b	20.17 ± 1.71 a	11.79 ± 6.66 a	7.13 ± 5.3 ab	20.2 ± 5.78 a
Glomus.acnaGlo2_VTX00155	3.58 ± 0.63 b	6.61 ± 4.42 b	6.71 ± 1.49 b	27.42 ± 4.17 a	12.83 ± 5.49 b	12.64 ± 3.68 b
Glomus.Yamato08.A1_VTX00100	54.49 ± 8.96 a	50.67 ± 17.75 a	19.04 ± 10.93 b	0.22 ± 0.17 b	0.05 ± 0.01 b	5.42 ± 4.54 b

**Table 5 plants-09-00901-t005:** The description of the six grafted citrus plants with different scion/rootstock genotypes.

Sample Name	Scion	Scion Genotype	Rootstock	Rootstock Genotype
Poncirus/Poncirus	Poncirus	Poncirus trifoliata	Poncirus	Poncirus trifoliata
Citrange/Citrange	Citrange	Citrus sinensis×Poncirus trifoliata	Citrange	Citrus sinensis×Poncirus trifoliata
Mandarin/Poncirus	Mandarin	Citrus reticulate	Poncirus	Poncirus trifoliata
Pummelo/Poncirus	Pummelo	Citrus grandis	Poncirus	Poncirus trifoliata
Orange/Poncirus	Newhall sweet orange	Citrus sinensis	Poncirus	Poncirus trifoliata
Orange/Citrange	Newhall sweet orange	Citrus sinensis	Citrange	Citrus sinensis×Poncirus trifoliata

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
