# Peer review of "Influence of Citrus Scion/Rootstock Genotypes on Arbuscular Mycorrhizal Community Composition under Controlled Environment Condition"

_plants, 2020, doi:10.3390/plants9070901_

Round 1

Reviewer 1 Report

Dear authors, I have read with pleasure your work and I think the article is interesting. But, there are some points that need to be addressed in order to improve the entire material.

L40-42 - add a reference

L47-49 - Rewrite please. You want to say about the reduced specificity of AMF for one host and that they can form symbiosis with multiple hosts?

L143-189 2.3. Citrus genotypes shape the AMF community composition

The entire subsection is confusing. You use PCoA for analysis, but you do not really explain it. You have only 18 points on it, and they present a lot of large differences inside each of the six combinations. As I saw in the Table S1, there are large differences between replicates of each combination in term of AMF diversity and specific value for each taxon. I suggest you to use a different approach, maybe another ordination like PCA, NMDS or any other that show better your results. Also, you have quite a lower variance explanation and I think that making 3 different PCoA`s for 2 combinations does not help. You can use these 2 by 2 combinations for clustering your data on first PCoA. Also, in the first graph you use PC1 and PC2, and in the last 3 ones PC1 and PC3. This you should explain in Material and methods section.

L202-204 - Explain the colors in figure caption. Which one is red, blue etc. It will help the figure to stand alone.

L217 - Discussion section

You need to add more references to this section.

L305-318 - it is better to make a table with your scion/rootstock combinations. It will be easier to read them. 

L305-309 - please rewrite. It is a little bit unclear.

You have grown seeds in sterile conditions, after that you transplanted them in sterile sand. After one year you have planted in greenhouse. In greenhouse - what type of soil did you used? Have you analyzed the soil in order to identify the native AMF community? How much space a plant have in order to determine the characteristics of its own rhizosphere? How many plants did you used in the entire experiment? Have you randomized your samples? 

L360-372 - add which packages you used for statistics, specific for each test.

L363-364 - pay attention to reference (Lin et al. 2012).

L370-372 - name the test used for significant differences. Duncan, Tukey etc.

Author Response

Response to reviewer 1:

Q1: L40-42 - add a reference

Response:Yes, three closely related references are added here. See in Line 39-43 in the new manuscript.

  1. Davies, F.; Albrigo, L. Citrus; Acribia, S.A.: Zaragoz, 1999.
  2. Rillig, M.C.; Aguilar‐Trigueros, C.A.; Camenzind, T.; Cavagnaro, T.R.; Degrune, F.; Hohmann, P.; Lammel, D.R.; Roy, J.; van der Heijden, M.G.; Yang, G. Why farmers should manage the arbuscular mycorrhizal symbiosis. New Phytol 2019, 1-5.
  3. Srivastava, A.; Singh, S.; Marathe, R. Organic citrus: soil fertility and plant nutrition. Journal of Sustainable Agriculture 2002, 19, 5-29.

Q2: L47-49 - Rewrite please. You want to say about the reduced specificity of AMF for one host and that they can form symbiosis with multiple hosts?

Response:Thanks for your kind suggestion. We want to say the contributions of different AMF species on plants were different. Thus, we rewrite this sentence as “In addition to these abiotic factors, AMF community is also impacted by host plants. Because of the contribution made by different mycorrhiza to plant is varied [24,25], the host plants are able to shape the AMF communities with their functional traits [26,27], and the effect of host plant on AMF communities could be observed even at genotype level [28]”. See in Line 48-52 in the revised manuscript.

Q3: L143-189 2.3. Citrus genotypes shape the AMF community composition

The entire subsection is confusing. You use PCoA for analysis, but you do not really explain it. You have only 18 points on it, and they present a lot of large differences inside each of the six combinations. As I saw in the Table S1, there are large differences between replicates of each combination in term of AMF diversity and specific value for each taxon. I suggest you to use a different approach, maybe another ordination like PCA, NMDS or any other that show better your results. Also, you have quite a lower variance explanation and I think that making 3 different PCoA`s for 2 combinations does not help. You can use these 2 by 2 combinations for clustering your data on first PCoA. Also, in the first graph you use PC1 and PC2, and in the last 3 ones PC1 and PC3. This you should explain in Material and methods section.

Response:Yes, we fully agree with your opinion. To be honest, we have performed both PCA and PCoA analyses, and we use three algorithms for PCoA analysis, including weighted unifrac distance, unweighted unifrac distance and bray curtis distance. Finally, the PCoA analysis based on Bray-Curtis distances is selected for publication according to its clear clustering of most samples in this study. According to your helpful suggestion, we revise this part as described below.

  1. We explain the PCoA results in detailsin the revised manuscript. (Line 151-164)
  2. The variance explanation of Orange/Citrange and Citrange/Citrange is very low, and the results of 2 by 2 combinations are not necessary for our main conclusion. Thus, we remove the Figure 3. A, B and further demonstrate the Figure 2 in details with the clustering of these 2 by 2 combinations.(Line 165-187)
  3. Because of the samples can’t be separated clearly in PC1 and PC2 with all methods mentioned above, we use PC1 and PC3 to illustrate the difference between samples in Figure. 3. The detailed method is added in Material and methods sectionas “Different dimensions of PC1, PC2 or PC3 were shown in the PCoA map according to the clustering of samples”. (Line 375-376)

Q4: L202-204 - Explain the colors in figure caption. Which one is red, blue etc. It will help the figure to stand alone.

Response:Thanks for your constructive suggestion. The colors in Figure 4 are described in the figure legend. (Line 200-204)

Q5: L217 - Discussion section

You need to add more references to this section.

Response:Yes, more closely related references (see below) are added to this section. Thank you. (Line 233, 239, 241, 243, 248, 256 and 292)

  1. Wang, P.; Shu, B.; Wang, Y.; Zhang, D.; Liu, J.; Xia, R. Diversity of arbuscular mycorrhizal fungi in red tangerine (Citrus reticulata Blanco) rootstock rhizospheric soils from hillside citrus orchards. Pedobiologia 2013, 56, 161-167.
  2. Faggioli, V.S.; Cabello, M.N.; Grilli, G.; Vasar, M.; Covacevich, F.; Öpik, M. Root colonizing and soil borne communities of arbuscular mycorrhizal fungi differ among soybean fields with contrasting historical land use. Agriculture, Ecosystems & Environment 2019, 269, 174-182.
  3. Saks, Ü.; Davison, J.; Öpik, M.; Vasar, M.; Moora, M.; Zobel, M. Root-colonizing and soil-borne communities of arbuscular mycorrhizal fungi in a temperate forest understorey. Botany 2014, 92, 277-285.
  4. Davison, J.; Moora, M.; Jairus, T.; Vasar, M.; Öpik, M.; Zobel, M. Hierarchical assembly rules in arbuscular mycorrhizal (AM) fungal communities. Soil Biology and Biochemistry 2016, 97, 63-70.
  5. Martínez‐García, L.B.; Richardson, S.J.; Tylianakis, J.M.; Peltzer, D.A.; Dickie, I.A. Host identity is a dominant driver of mycorrhizal fungal community composition during ecosystem development. New Phytol 2015, 205, 1565-1576.
  6. Krüger, C.; Kohout, P.; Janoušková, M.; Püschel, D.; Frouz, J.; Rydlová, J. Plant communities rather than soil properties structure arbuscular mycorrhizal fungal communities along primary succession on a mine spoil. Frontiers in microbiology 2017, 8, 719.
  7. Singh, G.; Mukerji, K.G. Root exudates as determinant of rhizospheric microbial biodiversity. In Microbial activity in the rhizoshere, Springer: 2006; pp. 39-53.
  8. Hugoni, M.; Luis, P.; Guyonnet, J.; el Zahar Haichar, F. Plant host habitat and root exudates shape fungal diversity. Mycorrhiza 2018, 28, 451-463.
  9.  Hazard, C.; Gosling, P.; Van Der Gast, C.J.; Mitchell, D.T.; Doohan, F.M.; Bending, G.D. The role of local environment and geographical distance in determining community composition of arbuscular mycorrhizal fungi at the landscape scale. The ISME journal 2013, 7, 498-508.
  10. Vieira, L.C.; Silva, D.K.A.d.; Escobar, I.E.C.; Silva, J.M.d.; Moura, I.A.d.; Oehl, F.; Silva, G.A.d. Changes in an Arbuscular Mycorrhizal Fungi Community Along an Environmental Gradient. Plants 2020, 9, 52
  11. Xu, X.; Chen, C.; Zhang, Z.; Sun, Z.; Chen, Y.; Jiang, J.; Shen, Z. The influence of environmental factors on communities of arbuscular mycorrhizal fungi associated with Chenopodium ambrosioides revealed by MiSeq sequencing investigation. Scientific Reports 2017, 7, 45134, doi:10.1038/srep45134.

Q6: L305-318 - it is better to make a table with your scion/rootstock combinations. It will be easier to read them. 

Response:Good suggestion. A new table with all the six scion/rootstock genotype combination is added in Material and methods section as Table 5. (Line 322)

Q7: L305-309 - please rewrite. It is a little bit unclear.

You have grown seeds in sterile conditions, after that you transplanted them in sterile sand. After one year you have planted in greenhouse. In greenhouse - what type of soil did you used? Have you analyzed the soil in order to identify the native AMF community? How much space a plant have in order to determine the characteristics of its own rhizosphere? How many plants did you used in the entire experiment? Have you randomized your samples? 

Response:Yes, the grafted citrus plants used in this study were generated in sterile conditions before transplanted into the experimental field.

Firstly, the soil in greenhouse is yellow soil, and all the soil in the experimental field was simply mixed by deep plowing for three times. Subsequently, five soil samples were randomly collected and soil properties were detected (see below), but no significant difference was observed.

Secondly, the native AMF community was not analyzed with high-throughput sequencing. Because AMF was different with microbes, the existing forms of microbes in root and rhizosphere were the same, but AMF was existed as spores in the rhizosphere.

Thirdly, a total of 18 individual grafted citrus plants were used in this study with three biological replicates for each genotype, and all the plants were randomly planted in the field in with 2m×2m space.

Sample

P (mg/kg)

K (mg/kg)

N (%)

Organic Matter (%)

pH

Soil 1

111.22

183.80

0.15

2.12

4.70

Soil 2

154.01

279.95

0.18

2.18

5.52

Soil 3

244.28

379.72

0.22

2.43

4.89

Soil 4

103.59

86.94

0.13

2.16

5.40

Soil 5

224.97

141.04

0.13

2.24

5.56

Q8: L360-372 - add which packages you used for statistics, specific for each test.

Response:Thanks for your suggestion. The packages for each test are clarified in the new manuscript. (Line 377-378)

Q9: L363-364 - pay attention to reference (Lin et al. 2012).

Response:We are so sorry for our mistake. This reference is corrected in the revised manuscript. (Line 368)

Q10: L370-372 - name the test used for significant differences. Duncan, Tukey etc.

Response:The test we used for detecting the differentially abundant AMF species was Fisher LSD test, and we have added it into the revised manuscript. (Line 377-378)

Reviewer 2 Report

6 out of 6 keywords: "arbuscular mycorrhizal fungi (AMF), citrus, community composition, scion, rootstock, genotype" are uninformative and redundant compared with the title "Influence of Citrus Scion/Rootstock Genotypes on Arbuscular Mycorrhizal Community Composition under Controlled Environment Condition"

lines 35-36 "c." unclear

Author Response

Response to Review 2

Q1: 6 out of 6 keywords: "arbuscular mycorrhizal fungi (AMF), citrus, community composition, scion, rootstock, genotype" are uninformative and redundant compared with the title "Influence of Citrus Scion/Rootstock Genotypes on Arbuscular Mycorrhizal Community Composition under Controlled Environment Condition"

Response: Thanks a lot. We revised the keywords as “AMF community composition, citrus, scion/rootstock genotype, Illumina Miseq sequencing, PCoA”. (Line 29-30)

Q2: lines 35-36 "c." unclear

Response: Yes, “c” is unclear to the readers. We revise it as “about” in the new manuscript. (Line 35-36)

Reviewer 3 Report

The manuscript entitled “Influence of Citrus Scion/Rootstock Genotypes on Arbuscular Mycorrhizal Community Composition under Controlled Environment Condition” presents interesting results about how the genotype of the rootstock and the scion could modify the presence Arbuscular mycorrhiza in grafted plants. The manuscript is interesting and easy to follow. Nevertheless, there are a few aspects to clarify in order to improve the overall quality of the article.

As a general comment, only Newhall sweet orange was grafted onto the two rootstocks, why the authors did not use the other scions grafted on the two different rootstocks? (i.e Pummelo/citrange or mandarin/citrange)

Line 100: please define VT at its first mention in the text

Line 149: “all the citrus root samples could be attributed to the genotypes” please clarify if it refers to the genotype of the rootstock

 Figure 2: could be possible to add Confidence ellipses?

Lines 158 311 312: please replace “trifoliate” by “trifoliata”

Lines 159 and 313: please replace “reticulate” by “reticulata”

Table 4: please indicate if the letters show differences within the same row or column

Author Response

Q1: As a general comment, only Newhall sweet orange was grafted onto the two rootstocks, why the authors did not use the other scions grafted on the two different rootstocks? (i.e Pummelo/citrange or mandarin/citrange)

Response: Thanks a lot for your comment. Indeed, samples were not compared one by one in the comparation analysis of AMF community composition. Further, there were many cultivars of citrus used as scions, but Newhall sweet orange was the most popular cultivar worldwide. Thus, in this study, we used the Newhall sweet orange as the common scion to illustrate the effect of rootstock genotypes on the AMF community composition under same scion.

Q2: Line 100: please define VT at its first mention in the text

Response: Yes. According to your suggestion, we revised this sentence as “A total of 475,917 tags (75.46% of the total clean tags) from 18 citrus root samples were annotated to 46 VT (virtual taxa, AMF molecular species) according to the MAARJAM database”. (Line 100-101)

Q3: Line 149: “all the citrus root samples could be attributed to the genotypes” please clarify if it refers to the genotype of the rootstock

Response: Sorry for the confusion. We revised it as “The PCoA analysis results revealed that the differences in AMF community composition among all the citrus root samples could be attributed to the genotype of citrus rootstock.” Thank you. (Line 150-151)

 Q4: Figure 2: could be possible to add Confidence ellipses?

Response: Thanks for your kind suggestion, confidence ellipses were widely used in PCoA analysis. However, the AMF community compositions of six different citrus scion/rootstock genotypes were investigated in this study, and samples with different scions grafted onto the same rootstock of Poncirus didn’t clustered together as one group. Thus, it is difficult to add the confidence ellipses here.

Q5: Lines 158 311 312: please replace “trifoliate” by “trifoliata”

Response: Yes, “trifoliate” were revised by “trifoliata” in the revised manuscript (Line 167-168 and 314-315). Additionally, we re-checked the name of citrus genotypes in whole manuscript to correct all possible mistakes.

Q6: Lines 159 and 313: please replace “reticulate” by “reticulata”

Response: According to your advice, “trifoliate” were revised by “trifoliata” in the revised manuscript. Thank you. (Line 169 and 316)

Q7: Table 4: please indicate if the letters show differences within the same row or column

Response: Thank you for your constructive suggestion. We added this information in the revised manuscript as “Different lowercase letters indicate significant difference within the same row (P< 0.05)”. (Line 214-216)

Round 2

Reviewer 1 Report

Dear authors, I have read with pleasure the new for of your manuscript and seems to be improved. It is now clear and easier to read and understand.